# A Lightweight Blockchain Scheme for a Secure Smart Dust IoT Environment

**Joonsuu Park and KeeHyun Park \***

Department of Computer Engineering, Keimyung University, Daegu 42601, Korea; joonsuupark@stu.kmu.ac.kr
\* Correspondence: khp@kmu.ac.kr; Tel.: +82-53-580-5266

**Abstract:** Since a smart dust Internet of Things (IoT) system includes a very large number of devices sometimes deployed in hard-access areas, it is very difficult to prevent security attacks and to alleviate bottleneck phenomena. In this paper, we propose a lightweight blockchain scheme that helps device authentication and data security in a secure smart dust IoT environment. To achieve our goals, (1) we propose the structure of the lightweight blockchain and the algorithm of processing the blockchain. In addition, (2) we reorganize the linear block structure of the conventional blockchain into the binary tree structure in such a way that the proposed blockchain is more efficient in a secure smart dust IoT environment. Experiments show that the proposed binary tree-structured lightweight blockchain scheme can greatly reduce the time required for smart dust device authentication, even taking into account the tree transformation overhead. Compared with the conventional linear-structured blockchain scheme, the proposed binary tree-structured lightweight blockchain scheme achieves performance improvement by up to 40% (10% in average) with respect to the authentication time.

**Keywords:** Internet of Things; blockchain; lightweight; security; smart dust; authentication

---

## 1. Introduction

Smart dust is the concept of smart dust which sprays dust-like tiny sensors on physical spaces such as buildings, roads, clothing, and the human body in order to detect information such as ambient temperature, humidity, acceleration, and pressure over the wireless network [1–3]. Smart dust is a network in which very small devices communicate organically, which can be thought of as an Internet of Things (IoT) [2–7] that deals with very small devices. That is, a smart dust IoT system can be considered as one of the special IoT systems dealing with smart dust devices which have very low computing power/resources and a very small size [1,2,8,9].

There have been many studies on security issues for IoT systems [10–13]. However, few studies have been done for the secure smart dust IoT system, although a smart dust IoT system is much more vulnerable to security attacks since smart dust devices have very limited computing power. For example, smart dust devices refer to devices that have very low communication speed and a small memory of less than 1 kb [14]. That is, the enormous amount of smart dust devices in a smart dust IoT environment can easily be exposed to widely known security attacks. In a smart dust IoT environment, where a very large number of devices collect data, unauthorized devices can interfere with data collection, resulting in data contamination.

In general, a blockchain is considered one of the good solutions to data reliability problems such as data forgery and tampering [15–17]. We introduced the concept of a blockchain into a smart dust IoT environment to solve the problems of authentication and data security. However, a blockchain constitutes linear chaining for the verification of ledgers, and unfortunately, the verification speed of ledgers is proportional to the number of nodes participating in the chain [18]. Since a smart dust IoT

environment assumes hundreds of millions of devices, it would experience a significant slowdown when using a conventional or pure blockchain scheme that has proof-of-work-based proof and linear chaining configuration.

Therefore, in this paper, we propose a lightweight blockchain scheme that helps device authentication and data security in a secure smart dust IoT environment. To achieve our goals, we reorganize the linear block structure of the conventional blockchain into a binary tree structure and lighten the blockchain operations to be efficient in a secure smart dust IoT environment. That is, we propose a simplified tree structure for Directed Acyclic Graph (DAG) and a tree transform structure required for the system. Experiments show that the proposed binary tree-structured lightweight blockchain scheme can greatly reduce the time required for smart dust device authentication, even taking into account the tree transformation overhead. Compared with the conventional linear-structured blockchain scheme, the proposed binary tree-structured lightweight blockchain scheme achieves performance improvement by up to 40% in the time required for authentication.

The remainder of this paper is structured as follows: Section 2 introduces the general concept of a smart dust IoT system and a blockchain. Section 3 introduces a lightweight blockchain for a smart dust IoT environment. Section 4 shows the experimental results that verified the validity of the proposed lightweight blockchain. Section 5 discusses the conclusion and future research.

## 2. Background and the Related Studies

### 2.1. A Smart Dust IoT System

A smart dust technology integrates technologies such as Microelectromechanical Systems (MEMS), an optical receiver, signal processing, and a control circuit, thick-film battery, and a solar cell [1–4,8,9]. One of the key features of smart dust device is that the devices are very tiny, though not nearly as small as dust. In a smart dust IoT systems, the enormous amount of smart dust devices can be sprayed using airborne devices (e.g., drones, aircraft, airplane, etc.) in areas that are difficult to access by most people (e.g., Amazon rain forest, lunar surface, polar regions, etc.). The nature of being deployed in hard-to-reach areas makes it difficult to follow up on the management of smart dust devices [3,9].

In a broad sense, a smart dust system can fall into the category of an IoT system. What makes a smart dust IoT system special is that it uses very tiny sensing devices with very low computing power/resources. The features of smart dust devices leave a variety of challenges such as bottlenecks, security, and device authentication in the smart dust IoT system. We proposed a framework of a smart dust IoT system to solve problems such as bottleneck removal and speed improvement through our previous studies [3,9].

Figure 1 shows the basic physical device configuration of a smart dust IoT system proposed in our earlier study [9]. As shown in the smart dust IoT system in Figure 1, the system is composed of multiple layers. SDDs (smart dust devices) in Figure 1 refer to sensing devices and have all the features of smart dust devices described above. RDDs (relay dust devices) are responsible for consolidating, transforming, and compressing data to remove bottlenecks, or as operating as a buffer, thereby distributing the load from a vast number of SDDs/RDDs. Processing Nodes are nodes that process the collected data and are mapped to AE (application entities) from a general IoT system perspective. Finally, the smart dust IoT server control and manage processing nodes.

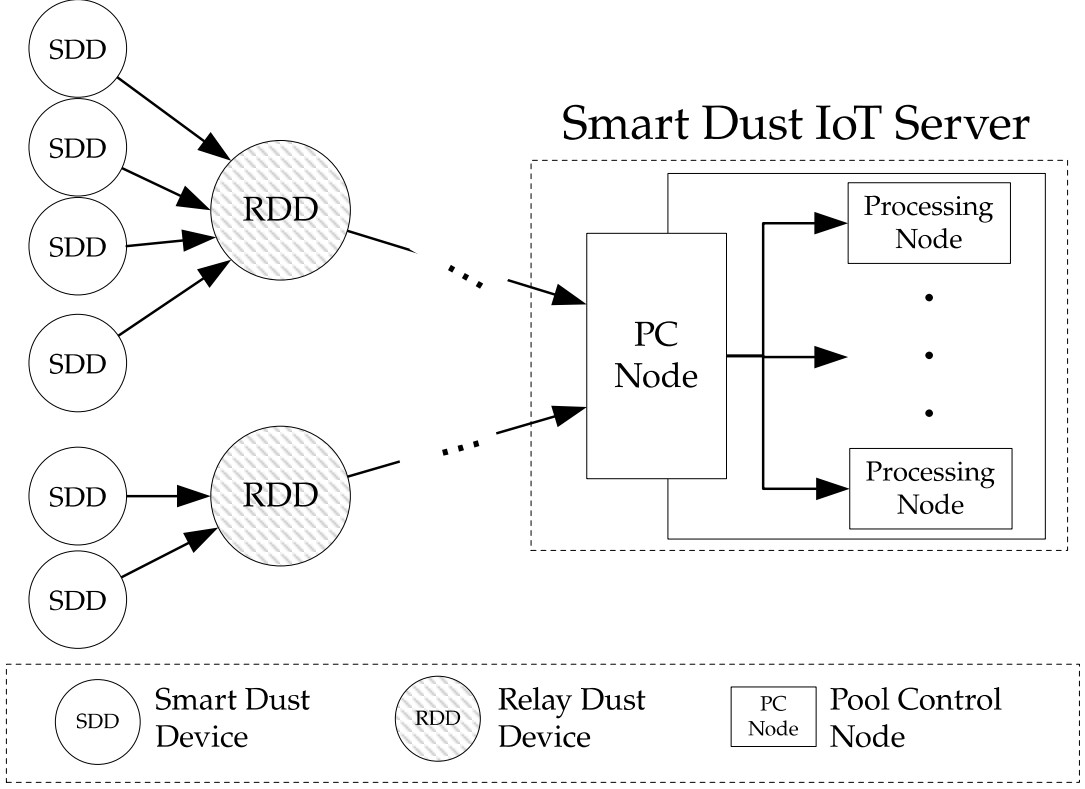

**Figure 1.** An overview of physical devices in a smart dust Internet of Things (IoT) environment [9].

*2.2. The Blockchain*

A blockchain is a technology based on distributed computing technology that connects the managed data to the data structure called "block" in the form of a hash chain so that no user can arbitrarily modify it, and anyone can view data (including the change history) [15–21]. A blockchain also refers to an algorithm that bundles multiple data transaction details (creation and change history) into blocks, connects multiple blocks like a chain using hashes, and then copies and stores them to be distributed by multiple people [15,16,21].

Figure 2 is an example of a generic block in a blockchain. Figure 2 contains only the minimum components to introduce the concept of a block. That is, concepts such as the Merkle tree root hash and mining [15,16,18] are omitted, and we only discuss how the integrity of ledgers is maintained. The most important point of the ledger integrity using blocks is that the block has a hash of the previous block (see Hash of the previous block in Figure 2).



**Figure 2.** An example of a generic block.

Figure 3 shows an example of a chain configuration using blocks (see Figure 2). As shown in Figure 3, each block has the hash value of the previous block, except for the Genesis Block (the block located in the first position), which is the block that does not have a previous block. When a block has a hash of the previous block, it means that in order to falsify or alter the specific data, the hashes of all blocks in front of it must be calculated and changed. The important point is that blocks are constantly being created while calculating the hashes of previous blocks. When a block is created, additionally, calculations are eventually increased. That is, it is impossible to change the entire ledger because blocks that must be constantly calculated are added even during an attack [15–21]. As can be seen intuitively in Figure 3, the time required for authentication increases dramatically as the number of nodes increase, due to the linear structure of a blockchain [20]. Therefore, it is very difficult or impossible to use a general blockchain in a smart dust IoT system with very limited computing power as well as a vast number of devices.

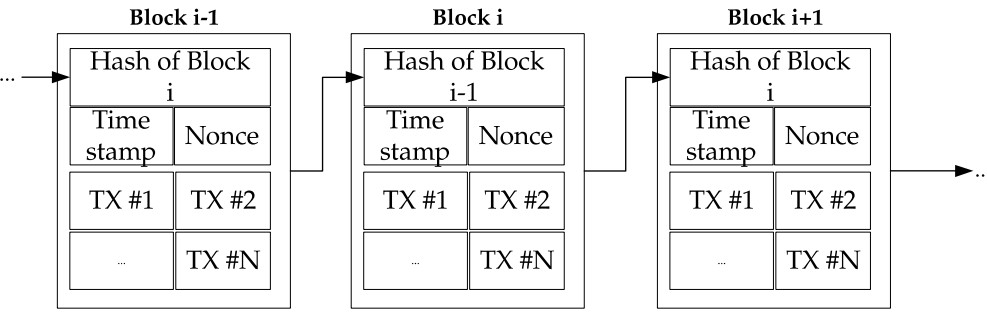

**Figure 3.** An example of chain configuration using blocks.

### 2.3. Blockchain for Resource-Constrained Environments

There are various previous studies that have considered IoT limited devices. [22] applies a lightweight hash function in consideration of limited devices in the IoT environment. Their main focus was to prioritize encryption performance, and performance evaluation also evaluates the simulation of the hash. In [23], the authors define and deal with the resources of IoT devices as necessary for calculations tailored to each system's purpose. They propose a system in [23] in which IoT devices lease resources required for computation from virtual resources. In addition, in [24], similar to [23], the resource required for the operation suitable for each system purpose is defined and treated as a resource. The difference between [24] and [23] is that it not only solves the resource problem, but also improves communication efficiency by using a new type of transaction and block, which reduces the amount of computation and storage overhead.

### 3. A Tree-Based Lightweight Blockchain for the Smart Dust IoT Environment

One of the well-known problems of chain algorithms included in the blockchain is slow processing speed [16,20]. Presently, the Bitcoin, the most well-known implementation of the blockchain, takes about 10 min to construct a new block [15,16], and in many cases, takes much more time to verify a block. The slow processing speed problem worsens as the number of nodes increases, which is inefficient for a smart dust IoT environment that encompasses so many devices.

The main features of a smart dust IoT system include a large number of devices and low computing power, which make it difficult for the normal blockchain to be used in a smart dust IoT environment. SDDs, with very limited computing power/resources, inherently cannot have ledgers for blockchains and the ability to perform operations such as comparing ledgers.

Table 1 shows a comparison between a linear-structured blockchain and a tree-structured blockchain.

**Table 1.** A comparison between a linear-structured blockchain [15,16,20] and a tree-structured blockchain.

| Conditions | Linear-Structured | Tree-Structured |
|---|---|---|
| Topology shape | Linear | Tree |
| Data propagation time | N | logN |
| Additional operations | Not needed | Needed |
| Implementation complexity | Low | High |
| Process speed | Slow | Fast |
| Fee | Needed | Not needed |
| Scalability | Low | High |

RDDs also do not have sufficient resources for the normal blockchain, though they are marginally improved than SDDs. Therefore, we need to make the blockchain light enough to run on RDDs. In order to improve the processing speed of the blockchain and to design a lightweight blockchain that can be used in the Smart Dust IoT environment, this study examined two considerations as follows:

- Simple authentication process considering RDD's computing power/resources
- Ledger validation performed at a faster rate than the normal linear blockchain

Figure 4 shows the lightweight blockchain for the smart dust IoT environments proposed in this study. In this figure, each block has six fields of information. The description of each field is as follows:

- Authentication Information: contains authentication results.
- Device's Information: has the device information of the SDD that has requested authentication.
- Prev. Hash: performs a chaining role (arrows in Figure 4) by having the SHA 256 hash information of the previous block.
- Hash (header): has the SHA 256 hash of fields including the header.
- Sync Time: has the synchronization time of the block that has set by the Time Node.
- Time Node is a node that determines the synchronization time (Sync Time) and is an external physical device.
- Transaction: has data sensed by SDDs.

When each device (SDD/RDD) of the system is deployed in the field, the authentication process is performed with the help of the Auth Node (Authentication Node). An SDD and an RDD are each authenticated by the related RDD and the SDD, respectively (See Section 3.1 for detailed descriptions).

We tried to keep the transaction as close to Ethereum [25] as possible, taking into account the possibility of future expansion. The difference is that there are "from" and "empty" fields. We wanted to reduce the amount of computation in consideration of low performance. Therefore, one can check "from" without calculation using the elliptic curve digital signature algorithm (ECDSA). Although this is a tradeoff between memory and computing power, we have considered a relatively expensive resource.

Table 2 shows fields of the transaction on proposed system. Among the transaction fields in Table 2, There is the sender of the message in the "From" field, and the "Empty" field is an extra field considering future expansion.

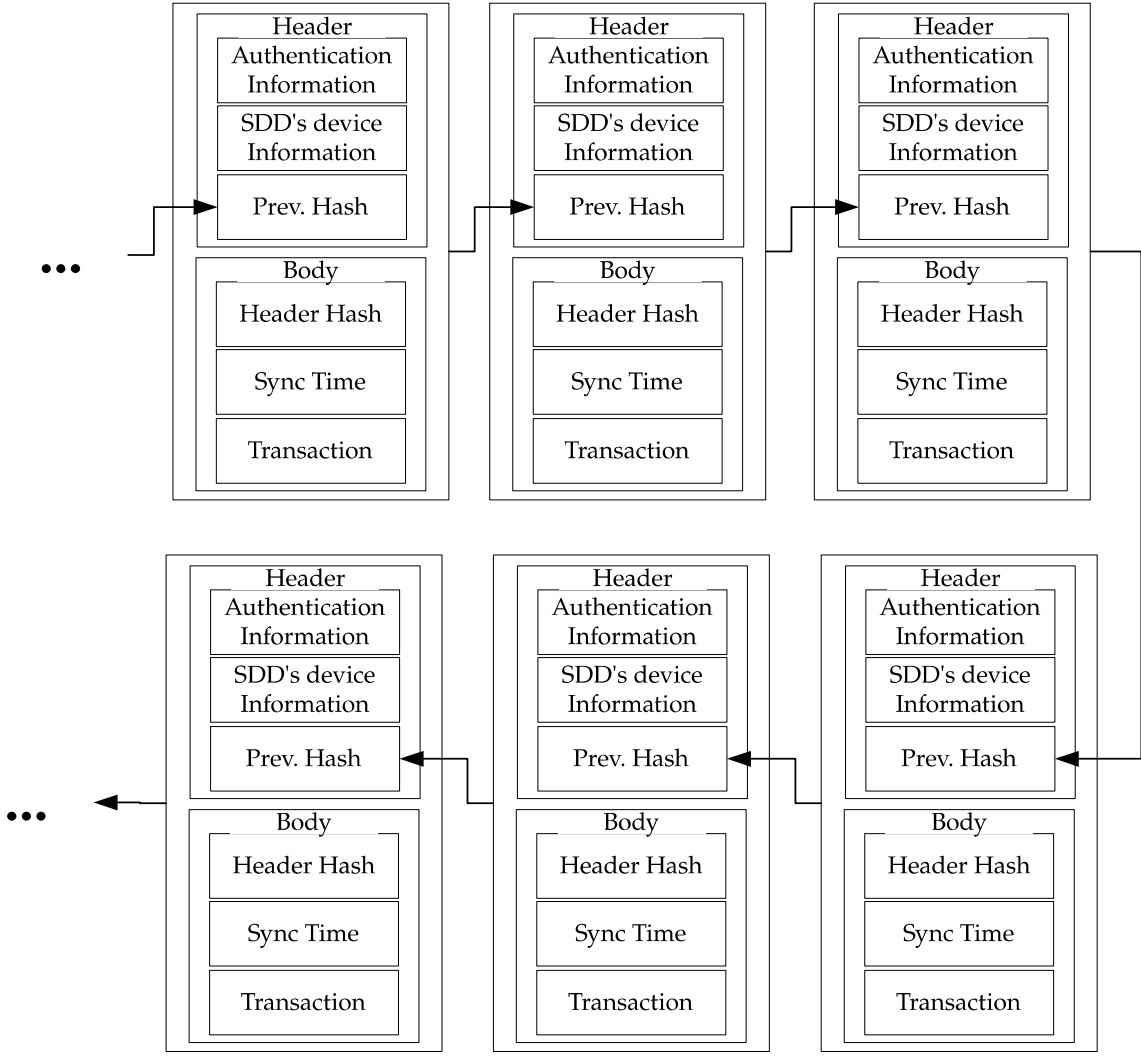

**Figure 4.** The lightweight blockchain proposed for smart dust IoT environments.

**Table 2.** Fields of transaction.

| 32 Bytes | 32 Bytes | 32 Bytes | 20 Bytes | 20 Bytes | 32 Bytes | 1 Bytes | 32 Bytes | 32 Bytes | 23 Bytes |
|---|---|---|---|---|---|---|---|---|---|
| Nonce | Price | Limit | To | From | Value | V | R | S | Empty |

Table 3 shows the authentication and block chaining process using the proposed lightweight blockchain. A detailed explanation for each step is described in Sections 3.1–3.5.

**Table 3.** The algorithm of the authentication and block chaining process using the proposed lightweight blockchain.

| Stages | Steps | Explanation |
|---|---|---|
| SDD/RDD [1] authentication (see Section 3.1) | Step 1 | An SDD sends an authentication request to RDD. |
| | Step 2 | The RDD authenticates the SDD with the help of the Auth Node. The RDD in turn sends an authentication response to the SDD. The SDD authenticates the RDD with the help of the Auth Node. |
| Creation and verification (see Section 3.2) | Step 3 | The RDD creates a new block. The RDD stores into the block the authentication information (result), the hash of the previous block, the SDD's device information, the sensed data by the SDD, and the SHA 256 hash value. |
| | Step 4 | The RDD verifies the block through the lightweight tree-structured ledger verification (see Section 3.2.1). |
| Scheduling (see Section 3.3) | Step 5 | The RDD requests a commit time to the Time node. |
| | Step 6 | On receiving the commit time from the Time node, the RDD stores the commit time to the Sync Time field in the block. |
| Propagation (see Section 3.4) | Step 7 | The RDD propagates the block (created through Step 3–Step 6) to other RDDs using the lightweight tree-structured ledger verification. |
| | Step 8 | Every RDD that receives the propagated block writes the block to its standby ledger. |
| Synchronization (see Section 3.5) | Step 9 | When the commit time has been reached, the Time node sends notifications to all the RDDs, and RDDs receiving the notification commit their standby ledgers to their commit ledgers. |

[1] SDD (Smart Dust Device), RDD (Relay Dust Device).

## 3.1. SDD/RDD Authentication Stage

In the authentication stage, an SDD and the related RDD authenticate each other with the help of the Auth Node. That is, each SDD is authenticated by an Auth Node, and RDD is an intermediary of authentication. We assume that an SDD and an RDD have the SDD's private key and the RDD's private key, respectively. The Auth Node has the public keys and the device information of all the SDDs/RDDs. Figure 5 shows the SDD/RDD authentication process. When an SDD sends an authentication request to its neighboring RDD (Message 1 in Figure 5), the RDD forwards it to the Auth Node (Message 2 in Figure 5) because the RDD does not have any SDD information. The ECC (elliptic curve cryptosystem) [26] with the SDD's private key is used for the encryption of these request messages. On receiving the request Message 2 from the RDD, the Auth Node decrypts the message using the public key of the SDD. Then, the Auth Node compares the SDD's device information obtained from Message 2 with the information stored in the Auth Node's database. If the information of the two devices is identical, Message 4 including the authentication result (ok), and the SDD's public key is sent to the RDD. The message is encrypted with the RDD's public key and hence the RDD can determine whether or not the SDD authentication is successful.

In turn, the RDD sends Message 6 for the SDD to authenticate the RDD. In order to authenticate the RDD, the SDD needs to compare the RDD's device information sent by the RDD itself and by the Auth Node. When the Auth Node sends Message 4, the RDD's device information stored in the Auth Node is included. The device information is encrypted with the SDD's public key and hence the RDD cannot alter the device information. If the comparison by the SDD matches, the RDD is authenticated successfully by the SDD.

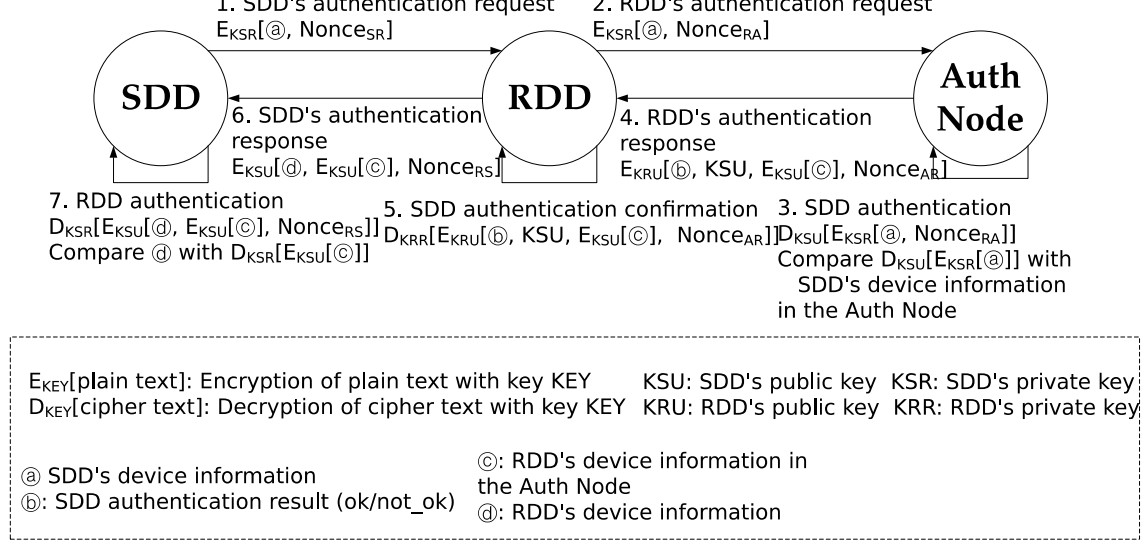

**Figure 5.** SDD/RDD authentication process.

## 3.2. Creation and Validation Stage

In the creation and validation stage, a new block is created and validated. When a new block is created, the RDD stores the related information in the new block. The Authentication Information field in the block contains the authentication result, authentication target, authentication time, etc. The Prev. Hash field, the Device's Information field, and the Transaction field of the block contain the hash of the previous block, SDD's device information, and the sensed data by the SDD, respectively. The RDD hashes the Header part of the block with the SHA 256 algorithm and stores it in the Header (hash) field.

Then, the validation step is performed. The RDD checks if the SDD has been authenticated using the lightweight ledger validation method proposed in this study. The lightweight ledger validation method, described in the next section, converts a linear-structured blockchain into a tree-structured blockchain and removes some unnecessary operations. It helps to quickly perform ledger validation, even if a large number of devices constitute the blockchain.

### 3.2.1. The Proposed Ledger Verification Method Using a Binary Tree-Structured Lightweight Blockchain

The unique linear chain structure of the conventional blockchain can be fatal in a system with a very large number of devices, such as a smart dust IoT environment. For example, more than 50,000,001 communications are required to validate a ledger of a smart dust IoT system with 100,000,000 devices, and all validation operations are "linear". Figure 6 is an example to demonstrate the problem caused by a linear chain structure. In this example, the shaded nodes are RDDs that participate in the ledger verification, and the unshaded nodes are the RDDs that do not participate in the ledger verification. If Node 1 requests the ledger validation, the validation process is performed linearly. That is, the process is repeated until more than half of the nodes confirm that the ledger is identical. In the example, a communication time of 50 ms is required to validate the ledger. If the time for Node 6 to notify the Node 1 is 10 ms, a total communication time of 60 ms is required.

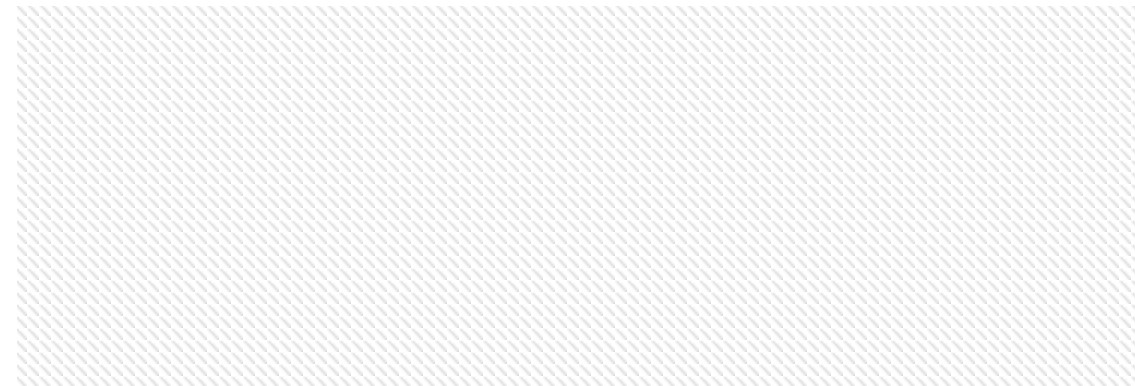

**Figure 6.** An example of a linear chain structure of the conventional blockchain.

Let us consider the proposed lightweight blockchain consisting of a binary-structured blockchain as shown in Figure 7. In the figure, the shaded nodes are RDDs that participate in the ledger validation, whereas the unshaded nodes are the RDDs that do not participate in the ledger validation. If Node 1 requests validation, Node 1 becomes the root node (RDD) of the binary tree and the root node requests validation to Node 2 and Node 3 in the tree structure. Since the validation processes at Node 2 and Node 3 can be performed in parallel, only 10 ms is required. In the next step, Node 2 and Node 3 request validation to Nodes 4, 5, 6, and 7 in parallel. Therefore, the total communication time required until more than half of the nodes participate in the validation is 20 ms. If the communication time for Node 6 to notify Node 1 is 10 ms, a total communication time of 30 ms is required, which is far less than that of the linear-structured blockchain mentioned above. If the examples shown in Figures 6 and 7 are increased to up to 1,000,000 device units, the linear-structured blockchain requires the communication time of 5,000,020 ms, whereas that of the binary tree-structured blockchain takes 230 ms.

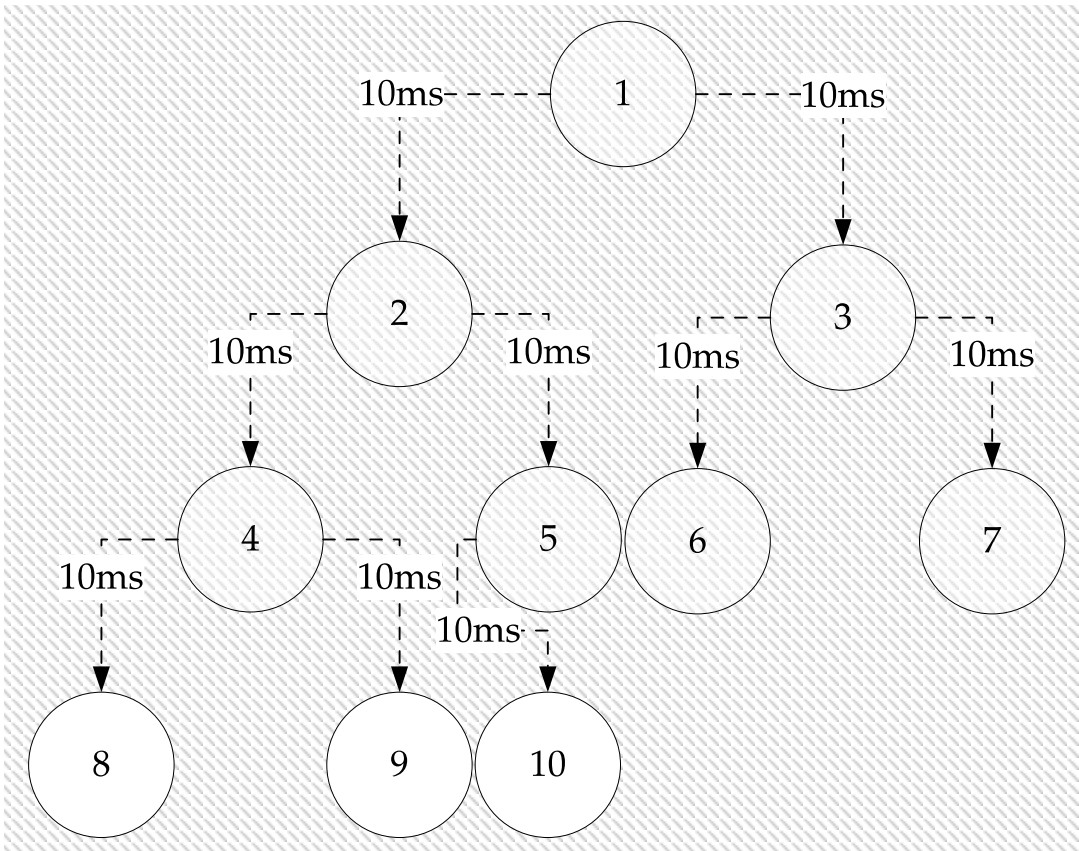

**Figure 7.** The proposed binary tree-structured lightweight blockchain.

The Auth Node designates the first RDD among all RDDs. The first RDD constructs a logical overlay network in a binary-structured fashion that connects all RDDs registered in the Auth Node. Each RDD also has its binary-structured overlay network. Except for the first RDD, the tree in each RDD may not have all RDDs. This means that each RDD has the subtree of the first RDD's tree. In addition, each RDD has a connection path to the first RDD. When a new RDD is connected, the first RDD adds the RDD to its logical overlay network. Then, the first RDD notifies the parent RDD of the new RDD that it has registered a child RDD, and the parent RDD adds the child RDD to its tree.

Table 4 shows the ledger validation algorithm using the binary tree-structured lightweight blockchain proposed in this study.

**Table 4.** The ledger verification algorithm using the binary tree-structured lightweight blockchain.

| Steps | Explanation |
| --- | --- |
| 1 | The RDD that wants its SDD verification requests other RDDs to compare the commit ledgers. |
| 2 | The RDD performs the tree transformation (see Section 3.2.2 for details). Make logical connection paths from the RDD's descent RDD to the first RDD temporally. |
| 3 | The RDD requests verification of the commit ledger from itself down to descent RDDs. |
| 4 | The descent RDDs send verification results to the RDD. |

### 3.2.2. An Example of the Ledger Verification Using the Binary Tree-Structured Lightweight Blockchain

The ledger verification occurs continuously whenever authentication is performed in order to prevent attacks through false authentication on devices. In addition, when collecting and transmitting sensed data, the ledger verification must be continuously performed to prevent duplicate data and data insertion attacks. This means that ledger verification occurs frequently, and that it can be requested from an arbitrary RDDs location. However, if the number of descendant RDD nodes is smaller than half of the total number of RDD nodes, verification cannot be performed. Another problem is that it is not possible to propagate locally generated blocks to ancestor RDD nodes due to the characteristic of a one-way tree structure. In order to solve these two problems, the tree structure must be flexibly changed whenever verification is requested.

Figure 8 shows a scenario for solving the above problems by transforming the tree structure. As shown in this figure, when an SDD requests registration (or authentication) to RDD D, only RDD D's descent RDD nodes (RDD H and RDD I) can participate in the verification, due to the characteristic of a one-way tree structure. This means that the number of RDDs that participate in the verification is smaller than five, half of the total number of RDDs.

The tree transformation of RDD described below is performed by the RDD requesting registration/change/verification of the ledger. In order to solve the above problem, we changed the structure so that each RDD has an overlay path to the first RDD. Using the overlay path, the first RDD was temporarily registered as a child node of the leaf RDD at the lowest level. Figure 9 shows an example of the transformation of the tree in Figure 8. The shaded part in Figure 9 indicates that RDD A (the first RDD) is connected temporarily as a child node of RDD H (the leaf node) at the lowest level. This does not actually change the tree. Rather, it is simply a temporary tree transformation (i.e., connection of the first RDD node as a child node of the leaf RDD H). Since the overhead of the tree transformation is constant, the overhead cost increases linearly as the number of verification requests increases. Since verification can be performed in parallel in the tree structure, we need to perform verification in at most $\log N/2$ times (N: the total number of RDD nodes).

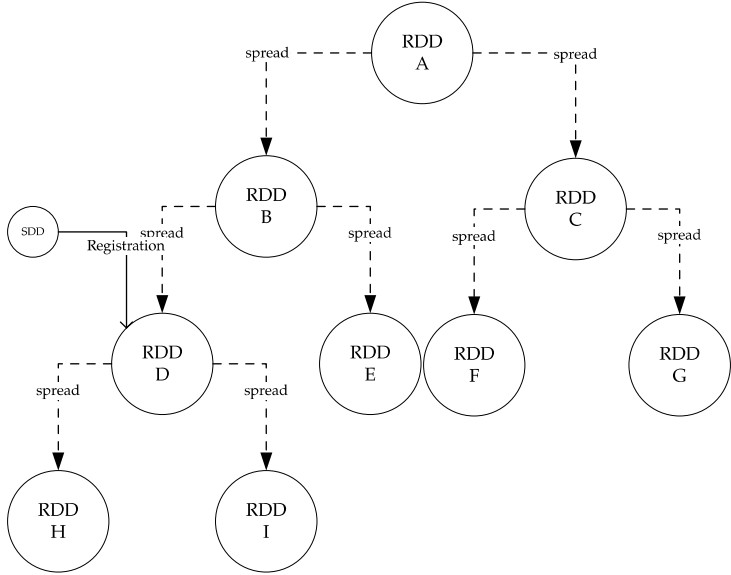

**Figure 8.** An example where more than half of the total number of RDDs cannot participate in validation.

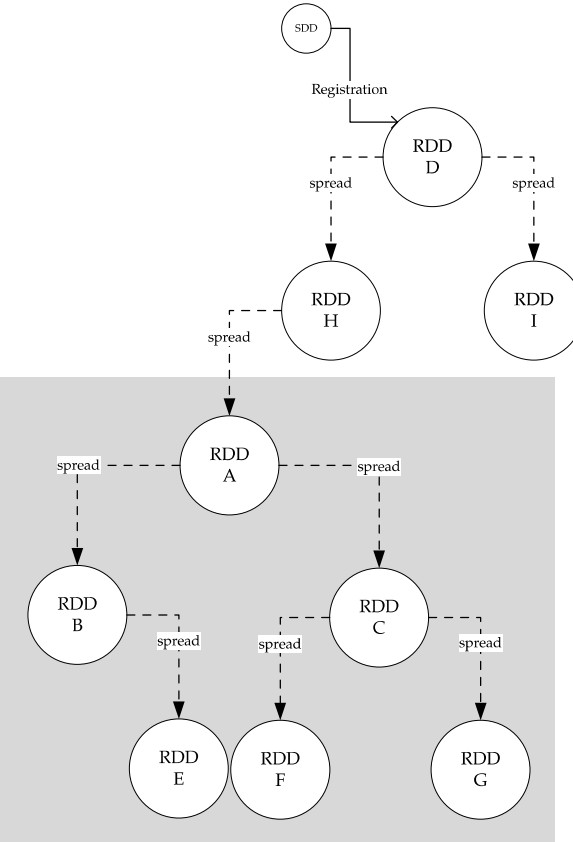

**Figure 9.** An example of the tree transformation in a case where more than half of the total RDDs cannot participate in validation.

*3.3. Scheduling Stage*

In the scheduling stage, the commit time for synchronization is determined. In order to synchronize the ledger, the conventional blockchain propagates the transaction, stores the transaction in the mempool, and sequentially applies it to the ledger. Therefore, conventional blockchain cannot validate transactions until the ledger is synchronized above a certain level. In addition, mempool is constantly

running in the background of the node, so it is difficult to use it in a smart dust IoT environment with limited computing power/resources.

Therefore, in this study, we propose a light synchronization method without using a mempool. To do this, we introduce three concepts to apply commit to the system: the standby ledger, the commit time, and the Time Node. The commit time is the time at which synchronization is performed as determined by the Time Node. The standby ledger is a temporary ledger in which transactions that have not been committed are stored because the commit time has not yet been reached. The commit ledger is the ledger of the blockchain to which the standby ledger is committed. The commit ledger matches to the ledger in the conventional blockchain. That is, all blocks are stored in the standby ledger above all, and when the commit time has reached, all the blocks in the standby ledger are committed to the commit ledger. Both ledgers types (standby/commit ledger) are owned by each RDD.

The Time Node determines the commit time and notifies the related RDDs that the commit time has been reached. The Time Node operates as a single thread, ensuring the order of transactions. This is a very simple but important process because it becomes a tool that can uniquely identify transactions. In the previous procedure (Section 3.2), after the RDD performs the ledger validation for the block, the RDD requests the Time Node to schedule a commit time. The scheduling is a simple task of recording the names of RDDs in an empty slot in a table divided by 3 min (enough time for the block to be committed, which of course can grow depending on the size of a block). Then, the time of the slot is returned to the RDD. The RDD records the slot time as the commit time in the Sync Time field in the block.

### 3.4. Propagation Stage

In the propagating stage, the RDD propagates the block created by the RDD to all other RDDs. After the RDD writes the commit time to the Sync Time field in the block, there should be no more data to be written to the block (i.e., the block is complete).

The block propagation process is very similar to the ledger validation process. The block propagation can use the same path used for the ledger validation. Therefore, the propagation uses a lightweight ledger validation using a tree structure. Finally, all the RDDs store blocks in their standby ledger.

### 3.5. Synchronization Stage

The final stage, synchronization of the ledger stage, is the stage at which every RDD commits its standby ledger to its commit ledger. When the commit time scheduled in the scheduling stage has been reached, the Time Node sends a notification to all the RDDs in the system. The RDDs receiving the notification commit their respective standby ledgers to the commit ledger.

### 4. Implementation and Experiments

Figure 10 shows the software module configuration for the binary tree-structured lightweight blockchain developed for this study. This system consists of SDD devices, RDD devices, the Auth Node server, and the Time Node server.

The software modules in common are Init Module, Session Module, Sending Module, and Receiving Module. The Init Module performs initialization according to the role of each device. The Session Module manages the communication connection to the device. The Sending Module sends data generated to other devices or servers. The Receiving Module receives data from other devices or servers.

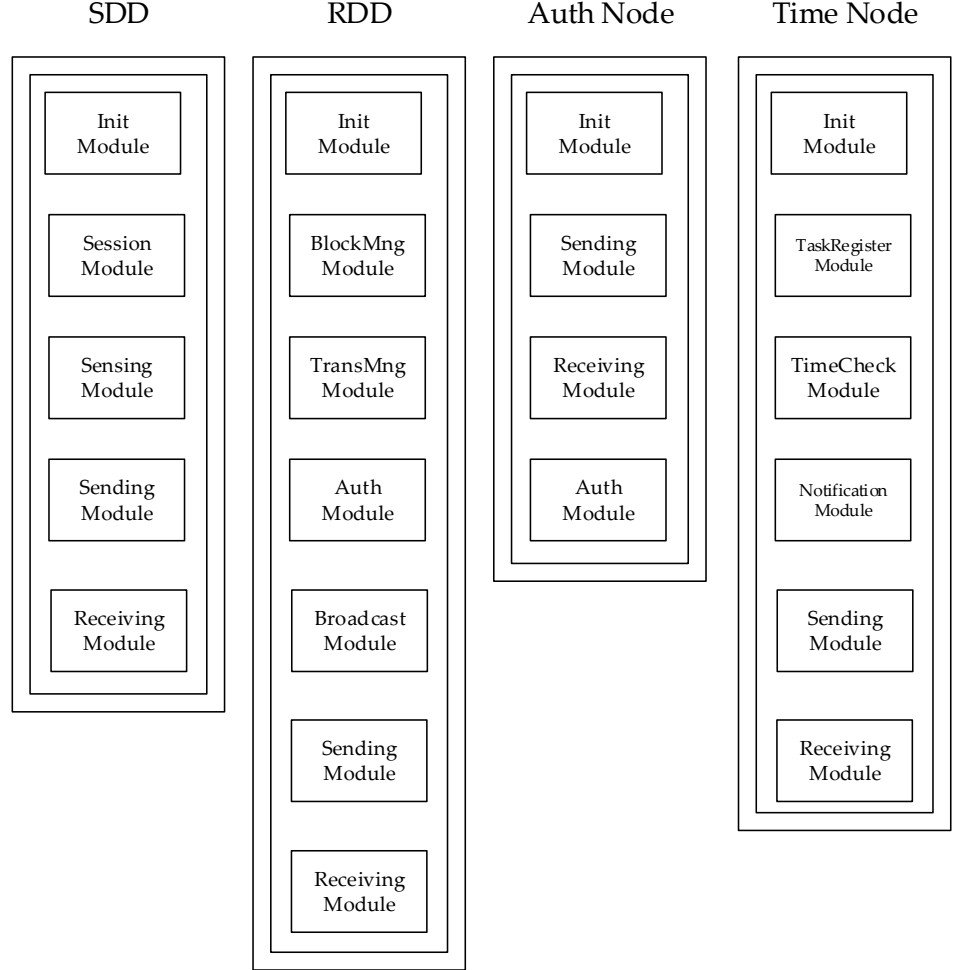

**Figure 10.** The software module configuration for the binary tree-structured lightweight blockchain.

The software modules in the SDD device are:

- Session Module: maintains and manages the connection between the device connected and this device
- Sensing Module: senses and collects data in the neighboring area

The software modules in the RDD device are:

- BlockMng Module: creates and manages blocks
- TransMng Module: records transactions
- Auth Module: performs authentication
- Broadcast Module: propagates the standby ledger

The software modules in the Auth Node server are:

- Auth Module: keeps authentication data in its database and manages authentication across the system

The software modules in the Time Node server are:

- TaskRegister Module: registers tasks in its schedule table
- TimeCheck Module: allocates synchronization time
- Notification Module: notifies when the synchronization time has reached

We also implemented the conventional linear-structured blockchain for comparison. Table 5 shows the experiment environments for this study.

**Table 5.** Experiment environments.

| Component | #1 | #2 | #3 |
|---|---|---|---|
| CPU | Intel i7-6700 | Intel i7-6700 | Intel i9-9900 |
| RAM | 16 GB | 32 GB | 64 GB |
| OS | Ubuntu 18.01 LTS | Ubuntu 18.01 LTS | Ubuntu 18.01 LTS |

We measured the authentication time required by gradually increasing the number of SDDs from 2000 to 7000. Ten percent of SDDs was assigned as RDDs. In addition, in consideration of the overhead required for tree transformation of the proposed tree structure lightweight blockchain, additional experiments were performed on the overhead time required, the total time required, and the ratio of the overhead. Figure 11 shows the total authentication time of all SDDs.

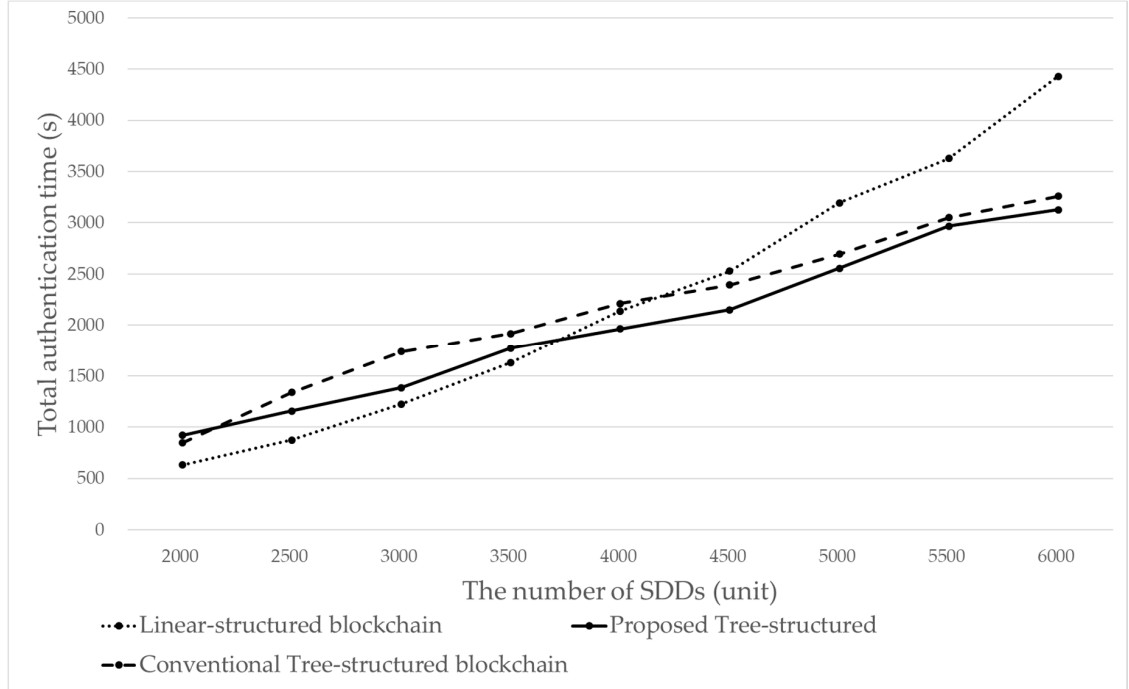

**Figure 11.** The total authentication time of all SDDs.

When the number of SDDs is 2000 to 3500, it can be seen that the conventional linear-structured blockchain provides better results than the binary tree-structured blockchain. However, when the number of SDDs exceeds about 3700, the performance of the binary tree-structured blockchain is better than that of the conventional linear-structured blockchain. Due to the limit of the experiment environments, we could not perform the comparison experiment beyond 6000 SDDs. However, with the trend of the two curves, we can predict the performance gap between the two blockchains is much wider beyond 6000 SDDs. This means that the binary tree-structured blockchain is more appropriate in a smart dust IoT system with an enormous amount of SDDs.

Compared with the conventional tree-structured blockchain (simple DAG-structured blockchain) and the linear-structured blockchain, the total authentication time of the proposed tree-structured blockchain is lower by about 7% and 10% on average, respectively. Since the only difference between the proposed tree-structured blockchain and the conventional tree-structured blockchain is the tree

transformation (Step 2 in Table 4), it can be said that the tree transformation operation contributes to the improvement.

Figure 12 shows the average authentication time per SDD. The average time required to authenticate one SDD for the binary tree-structured blockchain is almost constant (about 450 ms), whereas the average authentication time required for the conventional linear-structured blockchain continues to increase. This means that the conventional linear-structured blockchain would become more difficult or inefficient to use as the number of SDDs increases. It shows that the performance of the binary tree-structured blockchain is more improving than the linear chain as the number of SDDs increases.

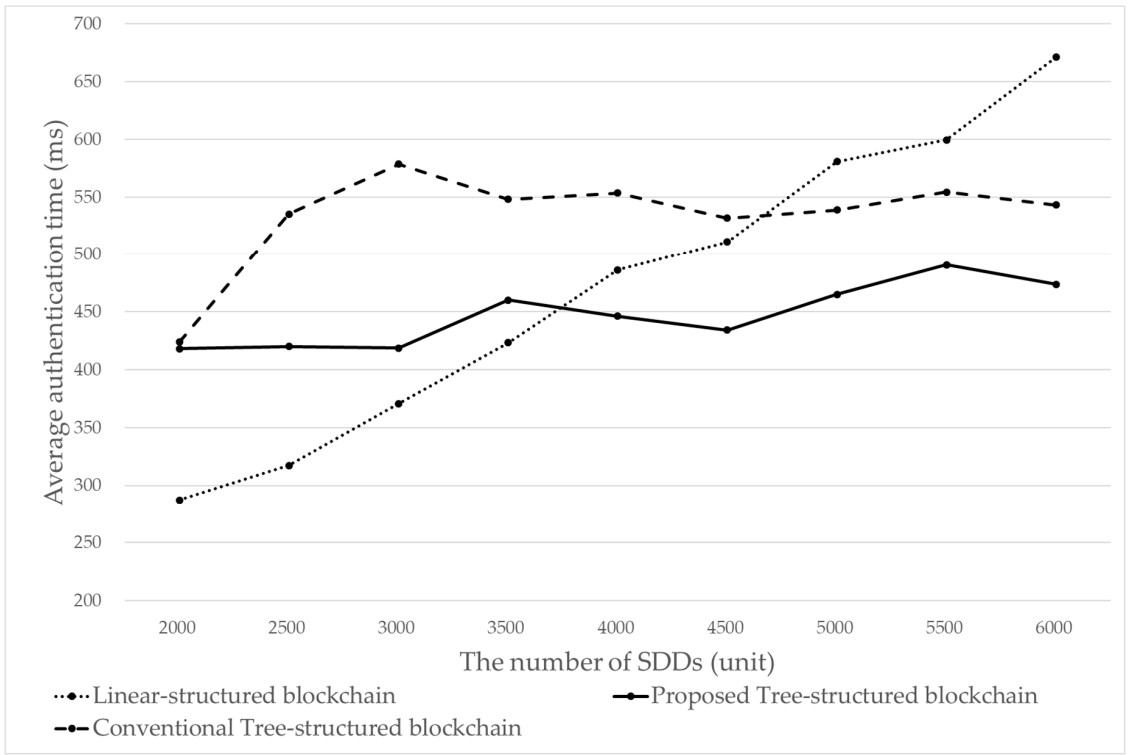

**Figure 12.** The average authentication time per SDD.

The conventional tree-structured blockchain does not show much difference from the proposed tree-structured blockchain when less than 2000 devices are connected, but shows an average performance difference of about 10% after more than 3000 devices are connected, and the gap cannot be narrowed.

Figure 13 shows the overhead time required for the tree transformation with regard to the number of SDDs. The overhead time required for the tree transformation increases as the number of SDDs increases. This is because the number of validations increase as the number of SDDs increases, which is very reasonable. What is considerable to note here is the ratio of the overhead time for the tree transformation to the total authentication time.

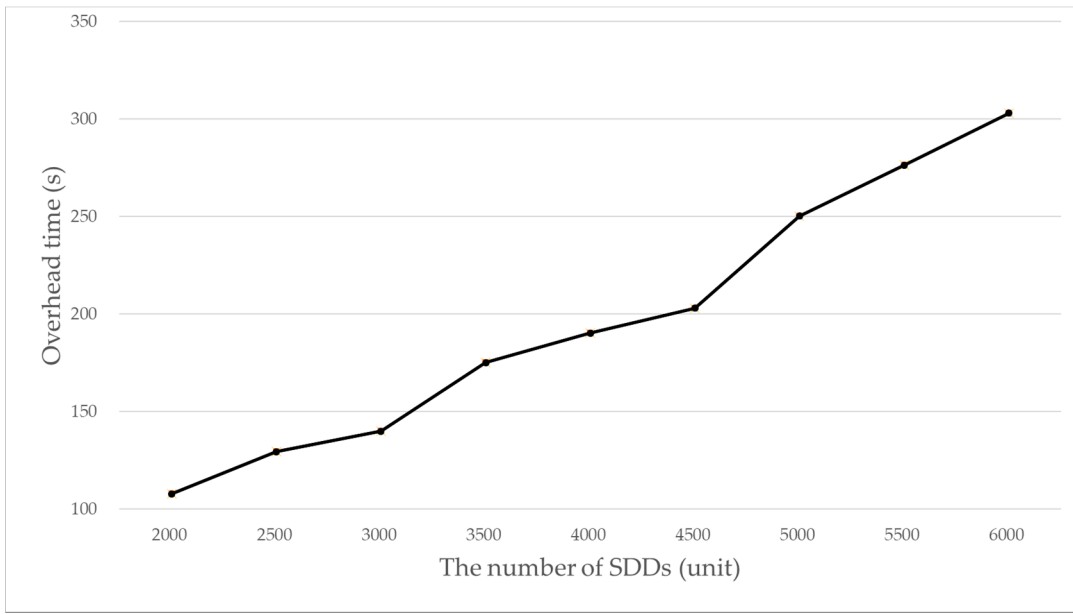

**Figure 13.** The overhead time required for the tree transformation.

Figure 14 shows the ratio of the tree transformation overhead time to the total authentication time. It can be seen that as the total number of SDDs increases, the ratio of the tree transformation overhead time to the total authentication time decreases and converges at about 10%. This also means that the binary tree-structured blockchain is more appropriate in a smart dust IoT system with an enormous amount of SDDs.

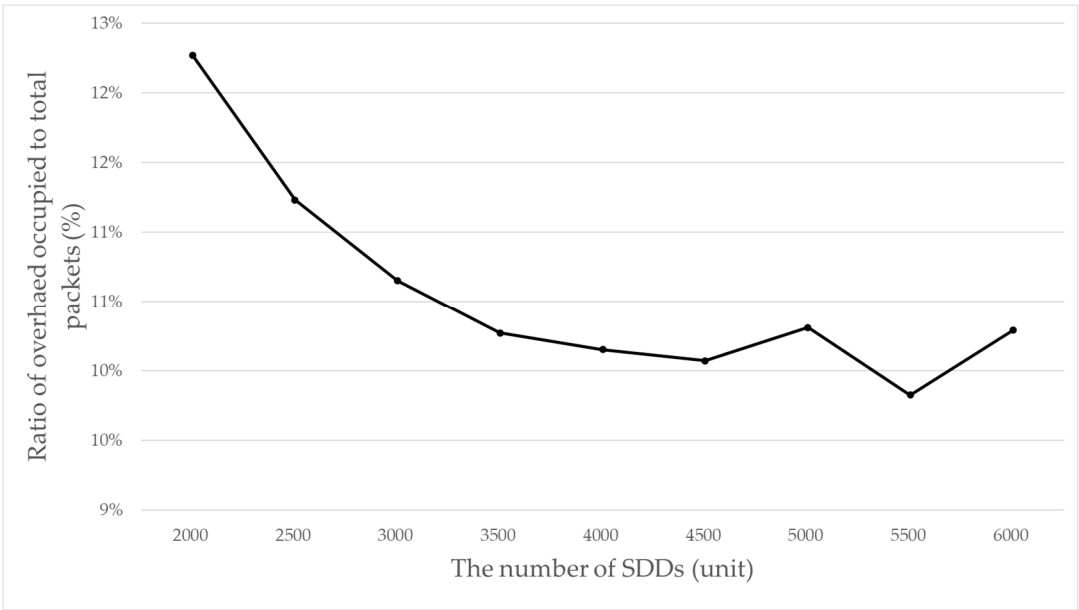

**Figure 14.** The ratio of the tree transformation overhead time to the total authentication time.

The results of the experiments mentioned above show that the proposed binary tree-structured lightweight blockchain can reduce the time required for SDD authentication, even taking into account the tree transformation overhead. Even the performance of the proposed blockchain is more improved compared to the conventional blockchain as the number of SDDs increases. The experimental results show that the performance improvement is up to 40% (10% in average) with respect to the authentication time.

## 5. Conclusions

Since a smart dust IoT system includes a very large number of devices sometimes deployed in hard-access areas, it is very difficult to prevent security attacks and to alleviate bottleneck phenomena. There have been many studies on security issues for IoT systems. However, few studies have been done for the secure smart dust IoT system, although a smart dust IoT system is much more vulnerable to security attacks since smart dust has a very limited computing power.

In this paper, we propose a lightweight blockchain scheme that helps device authentication and data security in a secure smart dust IoT environment. A smart dust IoT system includes a large number of devices and low computing power, which make it difficult for the normal blockchain to be used in a smart dust IoT environment. SDDs, with very limited computing power/resources, inherently cannot have ledgers for blockchains nor the ability to perform operations such as comparing ledgers. Therefore, we propose the structure of the lightweight blockchain and the algorithm of processing the blockchain. In addition, we reorganize the linear block structure of the conventional blockchain into the binary tree structure in such a way that the proposed blockchain is more efficient in a secure smart dust IoT environment. However, due to the characteristic of a one-way tree structure, if the number of the descendant RDD nodes in a binary tree-structured blockchain is smaller than half of the total number of RDD nodes, verification cannot be performed. In order to solve these two problems, the tree transformation is also proposed.

Experiments show that the proposed binary tree-structured lightweight blockchain scheme can greatly reduce the time required for smart dust device authentication, even taking into account the tree transformation overhead. Compared with the conventional linear-structured blockchain scheme, the proposed binary tree-structured lightweight blockchain scheme achieves performance improvement by up to 40% (10% in average) with respect to the authentication time.

**Author Contributions:** Conceptualization, K.P. and J.P.; methodology, K.P. and J.P.; software, J.P.; validation, K.P.; formal analysis, J.P.; investigation, K.P.; resources, J.P.; data curation, J.P.; writing—original draft preparation, J.P.; visualization, J.P.; review—editing and supervision, K.P.; project administration, K.P.; funding acquisition, K.P. All authors have read and agreed to the published version of the manuscript.

**Funding:** This research was funded by the Basic Science Research Programs through the National Research Foundation of Korea (NRF), grant number funded by the Ministry of Education, Science and Technology (No. NRF-2018R1D1A1B07043982).

**Conflicts of Interest:** The funders had no role in the design of the study; in the collection, analyses, or interpretation of data; in the writing of the manuscript, or in the decision to publish the results.

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
