# Peer review of "A Lightweight Blockchain Scheme for a Secure Smart Dust IoT Environment"

_applsci, doi:10.3390/app10248925_

Round 1
Reviewer 1 Report
In this paper, the authors propose and prototype a tree-structured blockchain for the authentication between IoT nodes. The proposed system is understandable and easy to follow, but its relation to smart dust is synthetic.
Major comments:
- According to the authors, the proposed system "helps device - authentication and data security." It is not clear why the system architect needs blockchain for that in the first place? Why no comparison with any conventional PKI system is given (it solves the mentioned problems out-of-the-shelf)? The discussion related to DAG is also missing.
- Why Smart Dust, introduced by Kristofer Pister, is referred to as 6(!!!) publications. The reviewer is sure that there was just one paper with an initial introduction. The paper should not be flooded with unnecessary or repetitive citations - only important ones should be kept to ensure the paper's readability.
- Related work section SHOULD reflect the works of IoT security and resource-constrained IoT device-related blockchain issues (e.g., search entry for "blockchain IoT resource-constrained devices") instead of one paragraph elaboration of what is smart dust. The current section appears to be an extended version of the introduction/background and should be either moved there or removed entirely.
- The performance evaluation campaign was executed on absolutely non-resource-constrained devices (i7). How were the performance limitations taken into account? Nonetheless, the results given in, e.g., Fig. 13, prove that the proposed system does not provide significant improvements compared to the linear structured one. It is not clear how the dust devices would benefit from it compared to some other approaches, e.g., PKI-based ones or DAG-based ones.
- Finally, it is not clear which node is responsible for tree transformation (how it is selected, how much computational efforts are required for its operation, etc.) and how to ensure the related connectivity and communication overheads.
Minor comments:
- Numerous punctuation and grammatical mistakes, e.g., a comma, are missing in the introduction's first sentence.
- "smart dust has very limited computing power" - smart dust does not have computational power since it is a paradigm - related IoT nodes have.
- References are broken from section 2.2.
- The orientation of flow in Figure 3 is inverted.
- Figure 5 represents a table, which should be added to the paper as a table.
- The university emails are a must for journal submissions to ensure the authors' universities' visibility.
Finally, the reviewer was unable to find any authors' relation to blockchain research in the past, which is required to propose a novel distributed security system. It is not clear how good the paper falls in the scope of the Applied Science journal. The authors may consider changing the focus from IoT dust authentication to a broader blockchain system design and submitting the paper to, e.g., a Cryptography journal.
Author Response
(Reviewer #1)
- According to the authors, the proposed system "helps device - authentication and data security." It is not clear why the system architect needs blockchain for that in the first place? Why no comparison with any conventional PKI system is given (it solves the mentioned problems out-of-the-shelf)? The discussion related to DAG is also missing.
Reply: The blockchain identifies assets through unique addresses. The PKI is used in the identification process. We do not propose an encryption method in this study. What we propose in this study is the tree topology and the translation method of the tree. Therefore, no comparison can be made. We described the authentication process using PKI in Section 3.1, by saying that “The Elliptic Curve Cryptosystem (ECC) [21] with the SDD’s private key is used for the encryption of these request messages.”
⇒ See: Lines 187 to 188 in Section 3.1.
We have performed experiments using the DAG-structured (conventional tree-structured) blockchain to compare with the proposed tree-structured blockchain with regard to authentication time.
⇒ See: Figure 11, Figure 12, Lines 384 to 389 and Lines 397 to 400 in Section 4.
- Why Smart Dust, introduced by Kristofer Pister, is referred to as 6(!!!) publications. The reviewer is sure that there was just one paper with an initial introduction. The paper should not be flooded with unnecessary or repetitive citations - only important ones should be kept to ensure the paper's readability.
Reply: We have deleted refs. [4, 5, 6].
- Related work section SHOULD reflect the works of IoT security and resource-constrained IoT device-related blockchain issues (e.g., search entry for "blockchain IoT resource-constrained devices") instead of one paragraph elaboration of what is smart dust. The current section appears to be an extended version of the introduction/background and should be either moved there or removed entirely.
Reply: Thank you. We have added Section 2.3 to discuss the related studies in resource-constrained environments. And we have changed the title of Section 2 to ‘Backgrounds and the Related Studies.’
⇒ See: Line 63 in Section 2, Lines 119 to 129 in Section 2.3.
- The performance evaluation campaign was executed on absolutely non-resource-constrained devices (i7). How were the performance limitations taken into account? Nonetheless, the results given in, e.g., Fig. 13, prove that the proposed system does not provide significant improvements compared to the linear structured one. It is not clear how the dust devices would benefit from it compared to some other approaches, e.g., PKI-based ones or DAG-based ones.
Reply: We have created a simulated environment where some resources are constrained by multiple processes (i.e., smart dust devices) working simultaneously. In this environment, three blockchain (i.e., the linear-structured blockchain, the proposed tree-structured blockchain, and the DAG-structured blockchain) systems were executed. Figure 13 (Figure 11, in the revised manuscript) shows that the performance of the proposed system provides better performance when more than 3,700 devices are connected. The figure shows that the proposed blockchain shows a performance increase of about 10% (up to about 40%) compared to the linear-structured blockchain, and about 7% (up to about 9%) compared to simple DAG-structured blockchain.
And, as mentioned earlier, the PKI method is not a comparison target in this study since the method is a part of the blockchain system.
⇒ See: Figure 11 and Figure 12 in Section 4.
- Finally, it is not clear which node is responsible for tree transformation (how it is selected, how much computational efforts are required for its operation, etc.) and how to ensure the related connectivity and communication overheads.
Reply: Thank you. On Step 2 in Table 4, “Perform the tree …” is changed to “The RDD performs the tree…”.
⇒ See: Step 2 in Table 4.
- Numerous punctuation and grammatical mistakes, e.g., a comma, are missing in the introduction's first sentence.
Reply: Thank you. We have corrected grammatical errors such as punctuation in this paper and asked a native English speaker for correction.
- "smart dust has very limited computing power" - smart dust does not have computational power since it is a paradigm - related IoT nodes have.
Reply: Thank you. “smart dust” is changed to “a smart dust device."
- Numerous punctuation and grammatical mistakes, e.g., a comma, are missing in the introduction's first sentence.
Reply: Thank you. We have corrected grammatical errors such as punctuation in this paper and asked a native English speaker for correction.
- "smart dust has very limited computing power" - smart dust does not have computational power since it is a paradigm - related IoT nodes have.
Reply: Thank you. “smart dust” is changed to “a smart dust device."
⇒ See: Line 35 in Section 1.
- References are broken from section 2.2.
Reply: Thank you. We used MS Word's cross-reference feature. However, the reference was broken during the editing process. We changed all references to plain text.
- The orientation of flow in Figure 3 is inverted.
Reply: Thank you. We wanted to highlight that the hash of block means the previous block. However, I think that this may cause confusion for readers because the direction is reversed to that of Figure 4. So, we changed the direction of the arrow in Figure 3.
⇒ See: Figure 3 in Section 2.
- Figure 5 represents a table, which should be added to the paper as a table.
Reply: We changed Figure 5/Figure 9 to Table 3/Table 4.
⇒ See: Table 3 and Table 4 in Section 2.
- The university emails are a must for journal submissions to ensure the authors' universities' visibility.
Reply: Thank you. We changed the private email address to the university email address.
⇒ See: Line 6
- Finally, the reviewer was unable to find any authors' relation to blockchain research in the past, which is required to propose a novel distributed security system. It is not clear how good the paper falls in the scope of the Applied Science journal. The authors may consider changing the focus from IoT dust authentication to a broader blockchain system design and submitting the paper to, e.g., a Cryptography journal.
Reply: Thank you. We have studied how to effectively apply the concept of the blockchain to IoT systems with many devices. That is, the main issue is to obtain a secure IoT system with a very large number of smart dust devices. Therefore, we have submitted this paper to an IoT-related journal. However, we will submit our next upcoming manuscript to the Cryptography journal.

Reviewer 2 Report
. In the abstract, “achieves performance improvement by up to 40%” in terms of what?
. There is over referencing in the article, e.g., [1-6] to reference the definition of smart dust? Similarly, [2, 6-9] to reference what? and [1-6, 10, 11]
. The problem with the linear structure and the advantage of the tree structure should be illustrated using diagrams.
. The related work is not a literature review, it is simply background section!
. There are many references or intent citations missing due to syntax errors!
. Figure 3 could be more expressive if you also update the transactions in each block.
. The slow processing speed in bitcoin is a design feature not due to the length of the chain! Why not only check the last block! (Section 3)
. Figure 5 is a table, I think it should be pseudocode.
. If a PKI is working smoothly, why do we need a blockchain? the dust networks are short lived and there are far more serious attacks on them than simply can be solved by authentication. Why the PKI is not enough to establish trust? How about masquerading attacks? What exactly is the additional value added by blockchain?
. When is the authentication done? Who is authenticating who? Why blockchain is a suitable option? What is the infrastructure to connect smart dust sensor networks to the blockchain?
To the best of my knowledge, smart dust concept never came to reality! I just checked the website of the inventors to this concept and they just sell standard IoT equipment!
Author Response
(Reviewer #2)
- In the abstract, “achieves performance improvement by up to 40%” in terms of what?
Reply: Thank you. To clarify the meaning of 40% in the abstract, we added: “with respect to the authentication time.” We also added this clarification in Section 4.
⇒ See: Lines 21 to 22 in the Abstract, Lines 423 to 424 in Section 4, Line 448 in Section 5.
- There is over referencing in the article, e.g., [1-6] to reference the definition of smart dust? Similarly, [2, 6-9] to reference what? and [1-6, 10, 11]
Reply: We have deleted refs. [4, 5, 6].
- The problem with the linear structure and the advantage of the tree structure should be illustrated using diagrams.
Reply: Thank you. We added Table 1 to summarize the comparisons between the two structures.
⇒ See: Lines 140 to 141 and Table 1 in Section 3.
- The related work is not a literature review, it is simply background section!
Reply: Thank you. We have added Section 2.3 to discuss the related studies in resource-constrained environments. And we have changed the title of Section 2 to ‘Backgrounds and the Related Studies.’
⇒ See: Line 63 in Section 2, Lines 119 to 129 in Section 2.3.
- There are many references or intent citations missing due to syntax errors!
Reply: Thank you. We used MS Word's cross-reference feature. However, the reference was broken during the editing process. We changed all references to plain text.
- Figure 3 could be more expressive if you also update the transactions in each block.
Reply: The contents of Figure 3 are not our proposed blocks; it is a visualization of blocks in a typical blockchain. The block of our proposed system can be seen in Figure 4. However, Figure 4 also does not mention transactions in detail because we were concerned about the picture becoming too complicated.
- The slow processing speed in bitcoin is a design feature not due to the length of the chain! Why not only check the last block! (Section 3)
Reply: Bitcoin's slow processing speed is due to the structural characteristics of the chain. However, this slowdown increases exponentially as the number of blocks and nodes increases. To solve this problem, the third generation blockchain tries to change the structure of the blockchain by trying various methods such as DAG.
The reason we did not only check the last block is that we are uncertain about the possibility of tampering with the last block.
- Figure 5 is a table, I think it should be pseudocode.
Reply: We changed Figure 5/Figure 9 to Table 3/Table 4.
⇒ See: Table 3 and Table 4 in Section 3.
- If a PKI is working smoothly, why do we need a blockchain? the dust networks are short lived and there are far more serious attacks on them than simply can be solved by authentication. Why the PKI is not enough to establish trust? How about masquerading attacks? What exactly is the additional value added by blockchain?
Reply: A PKI is used as a means of authenticating each node. This ensures that the creation and delivery of data are verified. However, verified creation and delivery do not guarantee that data will be stored/shared intact. Blockchain makes up for this. The PKI is used in the identification process in this study. We described the authentication process using PKI in Section 3.1, by saying that “The Elliptic Curve Cryptosystem (ECC) [21] with the SDD’s private key is used for the encryption of these request messages.”
- When is the authentication done? Who is authenticating who? Why blockchain is a suitable option? What is the infrastructure to connect smart dust sensor networks to the blockchain?
Reply: Thank you. To further clarify, we describe in Section 3.1 that an SDD is authenticated by the Auth Node. In this process, the RDD plays an intermediary role, and the RDD is also authenticated.
⇒ See: Lines 187 to 188 in Section 3.1.
- To the best of my knowledge, smart dust concept never came to reality! I just checked the website of the inventors to this concept and they just sell standard IoT equipment!
Reply: Most studies about smart dust are being conducted by simulation. One of the main reasons for using simulation is that the communications infrastructure is not supported well. In the near future, we expect (and hope) that, with the commercialization of 5G, smart dust will also change into an environment that better facilitates actual experimentation.

Reviewer 3 Report
This paper proposes a lightweight Blockchain (BC) scheme that should help device authentication and data security in dust iot environments.
The paper has somewhat fine potential although it also has some fatal technical and foundational errors that need to be solved.
Following are my comments to this manuscript:
1- When authors say "limited computing power", what is the definition of 'limited'? I found no examples or references regarding this in the manuscript.
2- When authors say "conventional or pure blockchain scheme", what is the definition of that? and what is an unconventional BC scheme?
>>>>> As I understood from the text, authors mean by conventional a Proof-of-Work (PoW) based BC, and they are proposing some non PoW-based BC consensus mechanisms.
Accordingly,
3- In section 2.1, the following devices are defined: SDD, RDD, Processing Node, Smart Dust IoT server. However, some new devices later appear in the proposed system without any definition, such as Time Node!
4- Authors excluded the Merkle trees from their proposal and discussion.
However, as the proposal appears to be a substitution proposal of PoW-based BCs, MTs should not be excluded as they are actually used to perform the TX verification. Hence, MT discussion and performance comparison are fundamental for sound scientific proposal validity..!
5-In Sec. 2.2, the authors claim (and build their whole research goal about): according to ref. [21], the time required for authentication increases dramatically as the number of nodes increase.
>>> As this claim is far from the actual facts proved in the literature, I checked the mentioned ref. and found that it does not present such claim. In fact, Authentication has not been mentioned even once in the ref.!!!
6- In Sec.3: Authors claim that one of the well-known problems is the slow processing speed, and that the BitCoin (which is a PoW-based BC) consuming 10 mins to confirm a block is a widely known problem.
>>> in fact, this is not (as represented in the manuscript) a problem in BC systems, BitCoin actually increase the difficulty of the puzzle so that miners (intentionally) consume 10 mins to produce a new block.
Hence, this is not a problem as much as a solution for the consistency of the ledger problem (which I think the manuscript claims to solve as well)
7- how does the data sensed by SDDs, which is put in the Transaction field of a new block, look like? how many? what's the size, the limit, etc.?
8- Authors interchangeably used the terms: verification, validation, confirmation, synchronization, approval, acceptance...!!! this is not accurate representation of the proposal, I believe authors need to be more accurate in using/defining such terminology.
9- In some places, it feels like each SDD is connected to one RDD, while in other places of the manuscript, it seems that the SDD has several RDD "neighbors" to which it is "related" (I'm using the authors' words here). What do you mean by related? and how does an SDD decide to connect to one rather than the other?
10- why steps 3 and 4 are replicated in Figure 5? and is it a figure or a table (same Q. applies to Fig.9)? step 3: when did RDD get the sensed data by the SDD?
11- Where did the authors get the info. that 50.000.001 comm.s are required when there are 100m devices (ref. needed)?
12- What do authors mean when they mention the participation in a ledger? is it the block addition to the distributed ledger or being a part of the network?
13-It is really not clear to me if authors are aiming to propose a verification scheme of the ledger or an authentication scheme for BC nodes (as claimed in the abstract and intro.) ?
14- The proposed Binary tree is simply a Directed Acyclic Graph. What is the novelty of the proposal?
15- Does the proposed Binary tree describe the network topology or the chain linking? The manuscript seems to propose a network topology.
For example, in section 3.3, authors say that the commit ledger (which is the final ledger saving new blocks) is a linear and "conventional" one hence the chain is not a tree..!
So the tree proposal is for the comm.s model of the BC? well, even in the evaluation section (4), the comparisons were not scientifically sound, because if we need to compare with the "conventional" BC, where all BC nodes are connected in a mesh topology with many neighbors. However, authors compared their proposal with a BC network whose nodes are connected linearly..!
Also, according to the BC definitions mentioned in sec. 2.2: BC is a type of DB not a physical network..!
16- what is the attack model that was discussed in sec. 3.2.2? the duplicate data problem is not an attack, it is just a problem that BC solves for P2P networks using consensus algorithms!
17- where is the standby ledger stored?
minor comments:
>> at the first appearance of a concept in the manuscript, write the abbreviation in brackets. e.g. Smart Dust Devices (SDD)
>> many errors in referencing tables and figures.
>> many language miner mistakes that need to be modified.
Author Response
(Reviewer #3)
- When authors say "limited computing power", what is the definition of 'limited'? I found no examples or references regarding this in the manuscript.
Reply: The “limited computing power” of smart dust devices is related to the concept of smart dust. Because they are physically very small, low-cost devices, they have relatively very limited computing power, battery life, and communication capabilities. This can be found in [1-2]. However, we decided to add more detail and further clarify the "limited computing power," which was added to this paper.
⇒ See: Lines 35 to 37 in Section 1.
- When authors say "conventional or pure blockchain scheme", what is the definition of that? and what is an unconventional BC scheme?
>>>>> As I understood from the text, authors mean by conventional a Proof-of-Work (PoW) based BC, and they are proposing some non-PoW-based BC consensus mechanisms.
Accordingly,
Reply: Thank you. Yes. The conventional blockchain refers to a blockchain that has PoW-based proof and has a linear chaining configuration. And, although it is not the term we used in this study, the "unconventional" blockchain can be a concept that includes nonlinear chaining constructs, proof of work, etc. To further clarify, we have added a statement about this.
⇒ See: Lines 47 to 48 in Section 1.
- In section 2.1, the following devices are defined: SDD, RDD, Processing Node, Smart Dust IoT server. However, some new devices later appear in the proposed system without any definition, such as Time Node!
Reply: Thank you. We have added a description of the Time Node to the module description.
⇒ See: Lines 160 to 161 in Section 3.
- However, as the proposal appears to be a substitution proposal of PoW-based BCs, MTs should not be excluded as they are actually used to perform the TX verification. Hence, MT discussion and performance comparison are fundamental for sound scientific proposal validity..!
Reply: We synchronize the order of TX for the purpose of verifying TX. This eliminates MT by using the method of comparing the entire ledger. The reason why this operation is possible is that the TX is very lightweight. This is not a procedure that is specific to this study. Some blockchains also use MT for the purpose of CRC for simple error verification.
- In Sec. 2.2, the authors claim (and build their whole research goal about): according to ref. [21], the time required for authentication increases dramatically as the number of nodes increase.
>>> As this claim is far from the actual facts proved in the literature, I checked the mentioned ref. and found that it does not present such claim. In fact, Authentication has not been mentioned even once in the ref.!!!
- In Sec.3: Authors claim that one of the well-known problems is the slow processing speed, and that the BitCoin (which is a PoW-based BC) consuming 10 mins to confirm a block is a widely known problem.
>>> in fact, this is not (as represented in the manuscript) a problem in BC systems, BitCoin actually increase the difficulty of the puzzle so that miners (intentionally) consume 10 mins to produce a new block.
Hence, this is not a problem as much as a solution for the consistency of the ledger problem (which I think the manuscript claims to solve as well)
Reply to Comments 5 and 6: This is contained in the GHOST protocol and the study in [21]. The study in [21] said that by utilizing the tree structure proposed by the GHOST protocol, there is a 40 times increase in block generation speed. Bitcoin's increasing difficulty is closely related to mining. Controlling the block generation rate by adjusting the mining difficulty is "intentional" by the administrator, but the structural slowdown is "unintended" by the administrator.
- how does the data sensed by SDDs, which is put in the Transaction field of a new block, look like? how many? what's the size, the limit, etc.?
Reply: We did not make any specific mention of the transaction. We believe that specific contents about transactions are necessary to facilitate better understanding of readers. So, we have added the size and shape of the transaction to the body.
⇒ See: Lines 169 to 179 in Section 3.
8- Authors interchangeably used the terms: verification, validation, confirmation, synchronization, approval, acceptance...!!! this is not accurate representation of the proposal, I believe authors need to be more accurate in using/defining such terminology.
Reply: Thank you very much for your opinion. We have unified the terms to "verification" and "synchronization."
- In some places, it feels like each SDD is connected to one RDD, while in other places of the manuscript, it seems that the SDD has several RDD "neighbors" to which it is "related" (I'm using the authors' words here). What do you mean by related? and how does an SDD decide to connect to one rather than the other?
Reply: A SDD is connected to the closest RDD with the highest signal strength.
- why steps 3 and 4 are replicated in Figure 5? and is it a figure or a table (same Q. applies to Fig.9)? step 3: when did RDD get the sensed data by the SDD?
Reply: Thank you. We checked again for mistakes similar to this in the paper.
We changed Figure 5/Figure 9 to Table 3/Table 4.
⇒ See: Table 3 and Table 4 in Section 3.
Table 3 (in the revised manuscript) explains the authentication process. If one want to map this process to content about sensed data processing, you can map the authentication request in "Step 1" to a sensed data processing request.
- Where did the authors get the info. that 50.000.001 comm.s are required when there are 100m devices (ref. needed)?
Reply: 50,000,001 in Section 3.2.1 is an example for illustration purposes only. We have suggested a figure in excess of 50% required for verification. That is, the figure is just over 50% of 100,000,000.
- What do authors mean when they mention the participation in a ledger? is it the block addition to the distributed ledger or being a part of the network?
- 13-It is really not clear to me if authors are aiming to propose a verification scheme of the ledger or an authentication scheme for BC nodes (as claimed in the abstract and intro.)?
- The proposed Binary tree is simply a Directed Acyclic Graph. What is the novelty of the proposal?
Reply to Comments 12-14: What we are proposing is a kind of DAG. Our main contribution is to design an LW blockchain for smart dust devices, and we use concepts such as DAG to achieve that purpose. In particular, one of our main contributions is the operation of changing the shape of the tree for the lightweight blockchain.
We think this may be not fully understood by readers because it is not explicitly stated in this paper. So, we added the system's contribution to the text.
⇒ See: Lines 52 to 54 in Section 1.
- Does the proposed Binary tree describe the network topology or the chain linking? The manuscript seems to propose a network topology.
For example, in section 3.3, authors say that the commit ledger (which is the final ledger saving new blocks) is a linear and "conventional" one hence the chain is not a tree...!
So the tree proposal is for the commas model of the BC? well, even in the evaluation section (4), the comparisons were not scientifically sound, because if we need to compare with the "conventional" BC, where all BC nodes are connected in a mesh topology with many neighbors. However, authors compared their proposal with a BC network whose nodes are connected linearly...!
Also, according to the BC definitions mentioned in sec. 2.2: BC is a type of DB not a physical network...!
Reply: As you have said, what we are proposing is topology, which includes tree operations and lightweight structure to construct the topology.
Both commit ledger/standby ledger are ledgers reflecting the tree.
Your view of the experiment is reasonable. So, we determined that we needed to provide additional experimental data that could clearly show our contribution. Therefore, additional experiments were conducted, and the results are attached.
⇒ See: Figure 11 and Figure 12 in Section 4.
16- what is the attack model that was discussed in sec. 3.2.2? the duplicate data problem is not an attack, it is just a problem that BC solves for P2P networks using consensus algorithms!
Reply: The attack model we discussed is a data forgery problem. Being able to prevent duplicate data is a side benefit.
17- where is the standby ledger stored?
Reply: The standby ledger is stored in each RDD. That is, all RDDs have a standby ledger. This is shown in Table 1 (Figure 5 of the previous manuscript). However, we have added this point to this paper to clarify for our readers.
⇒ See: Lines 311 to 312 in Section 3.3.
- minor comments:
>> at the first appearance of a concept in the manuscript, write the abbreviation in brackets. e.g. Smart Dust Devices (SDD)
>> many errors in referencing tables and figures.
>> many language miner mistakes that need to be modified.
Reply: Thank you. We have revised the manuscript to reflect your suggestions. In addition, we checked for other errors and asked an expert for corrections.

Round 2
Reviewer 1 Report
Thank you for the revision, most of the reviewer’s concerns were addressed. However, some minor issues still remain before the paper could be technically accepted. The reviewer highly encourages the authors to CHECK their submission right before the upload of the prepared pdf visually – most of the errors are easy to fix but may annoy the editors and/or reviewers.
- Second authors’ email is not official university email – should be changed.
- Size of Figures 2, 3, 5, 6 should be increased so that the smallest fontsize matches the one of the captions.
- Also, check that all figures are in vector format (you may export the plots this way as well).
- Are still broken after Figure 3!!!
- References are better to be added to Table 1 for the ease or readability.
- Table 3 should be moved to a standalone page.
Author Response
Dear Reviewers
We are very thankful to the editors and the anonymous reviewers for providing their valuable suggestions and comments to improve our article.
We have incorporated all the suggestions and comments in the revised version of our article and hope that the updated version will satisfy the reviewers’ observations.
Regards,
(Reviewer #1)
- Second authors’ email is not official university email – should be changed.
Reply: We apologize as this seems to have been a problem that occurred in the last revision. Our email address has been updated to reflect the university domain.
⇒ See: Line 7.
- Size of Figures 2, 3, 5, 6 should be increased so that the smallest fontsize matches the one of the captions.
Reply: We confirmed that the font sizes in Figures 2, 3, 5, and 6 were too small for readers, and increased the font size for readability. In addition, the readability of the legend in Figure 1 has been improved.
⇒ See: Figures 1, 2, 3, 5, and 6.
- Also, check that all figures are in vector format (you may export the plots this way as well).
Reply: We wrote the manuscript in MS Word. Unfortunately, MS Word does not support figure attachments in SVG format. However, we have used SVG format for the final papers (in PDF and on the website). If necessary, we will provide the SVG-formatted figures to the editor.
- Are still broken after Figure 3!!!
Reply: We checked Figure 3 once more. However, for us, Figure 3 is displayed normally. Therefore, we redraw Figure 3 completely.
- References are better to be added to Table 1 for the ease or readability.
Reply: We wrote the linear-structured column in Table 1 by referring to several papers. The corresponding references have been added in the caption of Table 1.
⇒ See: The caption of Table 1.
- Table 3 should be moved to a standalone page.
Reply: We confirmed that Table 3 is divided into 2 pages, and moved it to a standalone page.
⇒ See: Table 3.

Reviewer 2 Report
The authors have addressed my comments.
Author Response
Dear Editor/Reviewers
We are very thankful to the editors and the anonymous reviewers for providing their valuable suggestions and comments to improve our article.
We have incorporated all the suggestions and comments in the revised version of our article and hope that the updated version will satisfy the reviewers’ observations.
Regards,
Reviewer 3 Report
This reviewer thanks the authors for their constructive responses that clearly enhanced the submitted paper.
However, I still have the following comments:
**I still do not see the novelty and the contributions of this paper.
1-Blockchain is a distributed ledger, yet the authors still treats it as if it was a network topology. Accordingly, they propose a tree-like topology instead of the mesh topology and claim that they modified on the Blockchain structure.
2-Authors say in their response that "We synchronize the order of TX for the purpose of verifying TX. This eliminates MT by using the method of comparing the entire ledger."
This implies that the whole ledger (including all history of TXs) shall be searched every time a TX needs to be verified.
Accordingly, authors are claiming that their approach of validating a TX is better (IN ITS COMPLEXITY) than the Simplified Payment Verification (SPV). However, no proof was provided regarding this specific claim.
3-Authors response of "The study in [21] said that by utilizing the tree structure proposed by the GHOST protocol, there is a 40 times increase in block generation speed."
clarifies that there is no novelty in the proposed system since the protocol was already proposed in the literature.
Furthermore, Authors claim in their text that "it is very difficult or impossible to use a general blockchain in a smart dust IoT system
with very limited computing power as well as a vast number of devices". However, the linear ledger used in their proposal IS THE CONVENTIONAL BLOCKCHAIN they are claiming to modify. Authors proposal is to deploy the GHOST protocol in a dust IoT environment, which has a topology of a Tree, is not a modification on the blockchain.
The GHOST protocol is a competitor to the conventional Blockchain (PoW-based), as far as I can see, in terms of security. However, referring to the GHOST reference, GHOST performed worse than the conventional Bitcoin protocol (PoW-based) as long as the computational power of the adversary is less than gamma.
Even when the GHOST performed better, the difference is very low that makes me had to ask: So what about the Proof-of-Stake, the Proof-of-Authority, the Proof-of-Elapsed-Time, the DBFT, etc.?
as the authors are proposing a network topology (which they underlined in their responses), the validation experiments should compare the proposed tree-like topology to the MESH topology. Comparing the proposed tree-like topology to a linear blockchain does not make any sense to me..!
if the authors,, again, are proposing a DAG blockchain instead of the linear blockchain, then the technical details are not sound because their proposed blockchain is actually linear as well (as clarified in the text) while THIS LINEAR CHAIN IS SAVED IN A TREE-LIKE NETWORK.
Finally: it is still not clear to me why authors want to add the Blockchain to their system anyway..! they could simply use a lightweight DB which does not imply consensus. The deployment of the BC should be justified each time it is deployed because it is proven that BC is much time and energy consuming than a classical DB.
Author Response
Dear Reviewers
We are very thankful to the editors and the anonymous reviewers for providing their valuable suggestions and comments to improve our article.
We have incorporated all the suggestions and comments in the revised version of our article and hope that the updated version will satisfy the reviewers’ observations.
Regards,
(Reviewer #3)
1. Blockchain is a distributed ledger, yet the authors still treat it as if it was a network topology. Accordingly, they propose a tree-like topology instead of the mesh topology and claim that they modified on the Blockchain structure.
Reply: It is true that a very large part of what we are proposing is topology. As you have stated, blockchain is a distributed ledger. However, this does not only pertain to distributed ledgers. In general, blockchain is a collective term for a distributed ledger and the accompanying technologies for it. I believe our paper needs to be interpreted in a broader sense as it includes not only topology but also actions for removing the concept of mining.
2. Authors say in their response that "We synchronize the order of TX for the purpose of verifying TX. This eliminates MT by using the method of comparing the entire ledger."
This implies that the whole ledger (including all history of TXs) shall be searched every time a TX needs to be verified.
Accordingly, authors are claiming that their approach of validating a TX is better (IN ITS COMPLEXITY) than the Simplified Payment Verification (SPV). However, no proof was provided regarding this specific claim.
Reply: SPV requests TX information from the full node for TX verification. At this time, the full node uses the Merkle tree to verify the TX. The nodes proposed in this paper do not need to use Merkle Tree for TX verification because all nodes can run like full nodes. This is what differentiates our system from SPV.
3. Authors response of "The study in [21] said that by utilizing the tree structure proposed by the GHOST protocol, there is a 40 times increase in block generation speed."
clarifies that there is no novelty in the proposed system since the protocol was already proposed in the literature.
Furthermore, Authors claim in their text that "it is very difficult or impossible to use a general blockchain in a smart dust IoT system with very limited computing power as well as a vast number of devices". However, the linear ledger used in their proposal IS THE CONVENTIONAL BLOCKCHAIN they are claiming to modify. Authors proposal is to deploy the GHOST protocol in a dust IoT environment, which has a topology of a Tree, is not a modification on the blockchain.
The GHOST protocol is a competitor to the conventional Blockchain (PoW-based), as far as I can see, in terms of security. However, referring to the GHOST reference, GHOST performed worse than the conventional Bitcoin protocol (PoW-based) as long as the computational power of the adversary is less than gamma.
Even when the GHOST performed better, the difference is very low that makes me had to ask: So what about the Proof-of-Stake, the Proof-of-Authority, the Proof-of-Elapsed-Time, the DBFT, etc.?
as the authors are proposing a network topology (which they underlined in their responses), the validation experiments should compare the proposed tree-like topology to the MESH topology. Comparing the proposed tree-like topology to a linear blockchain does not make any sense to me..!
if the authors,, again, are proposing a DAG blockchain instead of the linear blockchain, then the technical details are not sound because their proposed blockchain is actually linear as well (as clarified in the text) while THIS LINEAR CHAIN IS SAVED IN A TREE-LIKE NETWORK.
Reply: We cite a study that the GHOST protocol can achieve 40 times the performance in very ideal cases. This is cited for the purpose of suggesting that the configuration of the node and the shape of the topology can be an important issue for improving performance.
The system we are dealing with reduces the amount of computation by excluding the inherited block weights of the GHOST protocol. We propose an action like tree transformation for this. This is a very important factor in our system as our target is SMART DUST DEVICES.
Several studies have proven that the GHOST protocol can solve the problems of blockchain. Yonatan Sompolinsky's paper “Secure High-Rate Transaction Processing in Bitcoin” is a representative study.
PoS gave us an important idea. The part related to the consensus algorithm is considered an open issue in our paper. However, we are preparing a consensus algorithm in a mixed form of PoW and PoS in our next paper.
We tried to show the basis for our arguments through experiments. The key was whether we could show that the changes we made in the same environment would be more efficient. However, constructing various overlay networks on top of various physical environments is the direction we will expand in the future.
4. Finally: it is still not clear to me why authors want to add the Blockchain to their system anyway..! they could simply use a lightweight DB which does not imply consensus. The deployment of the BC should be justified each time it is deployed because it is proven that BC is much time and energy consuming than a classical DB.
Reply: We use the blockchain for the following purposes:
- Invariant of node collection data
- Two-factor authentication/Absence of duplicate data
- Eliminate the load concentrated on the database
These are the same concepts as blockchain's decentralization and double payment prevention.
Technically, I believe this problem can be solved if the DB is designed very well. However, this is not an easy problem. Also, for other blockchain systems, blockchain may not be necessary if the system is very robust.
